# The therapeutic effect of glycyrrhizic acid compound ointment on imiquimod-induced psoriasis-like disease in mice

Yanwen Zhang[1], Qian Wang[1], Shuangyong Sun [1]*, Lingyan Jiang[2]*

**1** School of Pharmacy, Tianjin Medical University, Tianjin, China, **2** TEDA Institute of Biological Sciences and Biotechnology, Nankai University, Tianjin, China

* ewen1122@qq.com (SS); wang_qq1999@163.com (LJ)

**Data Availability Statement:** The data underlying the results presented in the study are available

## Abstract

Glycyrrhetinic acid, a drug with anti-inflammatory effects, enhanced the activity of antipsoriatic efficacy. In this research, an ointment with glycyrrhetinic acid was prepaired as the major component and several other herbal monomers (astilbin, osthole, and momordin Ic) have antipsoriatic activity as minor components. Then an Imiquimod-induced psoriasis-like mouse model was established and the damaged skin condition of the administered group, the changes in the spleen index and the secretion of inflammatory factors in mouse skin were observed. Calcipotriol ointment was used as a positive control to compare the efficacy. Glycyrrhizic acid compound ointment significantly improved imiquimod-induced psoriasis in mice and reduced the secretion of TNF-α, IL-12, IL-17, and IL-23 in mouse skin, and showed a stronger therapeutic effect than calcipotriol ointment. Calcipotriol ointment did not significantly alleviate imiquimod-induced splenomegaly and did not significantly reduce the expression of IL-17 and IL-23 in mouse skin. Glycyrrhetinic acid compound ointment was more effective than calcipotriol and was dose-dependent in the treatment of imiquimod-induced psoriatic dermatitis in mice. Meanwhile, calcipotriol was not suitable for the treatment of Imiquimod-induced psoriasis-like mice.

## Introduction

Psoriasis is a common chronic skin disease with a prevalence of 0.6%-4.8% [1]. Although the etiology of psoriasis is unknown, environmental factors (such as stress, illness, trauma, and drugs) and genetic vulnerability contribute [2], leading to inflammatory in which the skin's elevated production of pro-inflammatory cytokines and chemokines attracts immune cells, resulting in the proliferation of localized and invasive cells [3]. Psoriasis is caused primarily by pro-inflammatory cytokines such as interleukin 6 (IL-6), interleukin 1β (IL-1β), and tumor necrosis factor-alpha (TNF-α) from stressed keratin-forming cells. These cytokines stimulate plasmacytoid dendritic cell synthesis and interferon-α (IFN-α) release. IFN-α stimulates dendritic cells in the dermal bone marrow, prompting them to move to nearby lymph nodes and release IL-12 and IL-23. And then, IL-12 and IL-23 activate circulating T helper cells (types 1,

from Figshare. DOI: 10.6084/m9.figshare.22822346.

**Funding:** This study was supported by The National Natural Science Foundation of China (Grant No. 32170110). The funders had no role in study design, data collection and analysis, decision to publish, or preparation of the manuscript. All authors did not receive a salary from any of our funders.

**Competing interests:** The authors have declared that no competing interests exist.

17, and 22). These lymphocytes subsequently move to the skin and release IFN-γ, IL-17A, IL-17F, and IL-22, partaking in a complicated immunological interplay between local and invading cells, resulting in uncontrolled inflammation and keratin-forming cell overproliferation [4].

18β-Glycyrrhizinic acid (GA) is one of the main components of *Glycyrrhiza glabra* L., which has various pharmacological effects such as antioxidant, antitumor, and anti-inflammatory [5]. Recently, it has been demonstrated that glycyrrhizic acid mediates apoptosis of HaCaT keratin-forming cells [6] and enhances the antipsoriatic activity of some drugs [7]. The transdermal efficacy of glycyrrhetinic acid has not been demonstrated, and synergistic effects of glycyrrhetinic acid with other natural drug monomers have yet to be explored. In this study, we selected Chinese medicine monomer with previously reported antipsoriatic activity [8]. Then we made an ointment with glycyrrhizinic acid as the main ingredient and astilbin, osthole, and momordin Ic as minor ingredients. The therapeutic effect of the compound ointment on imiquimod-induced psoriasis-like dermatitis in mice was observed to explore its intervention effect and mechanism on psoriasis and to provide an experimental basis for the clinical application.

## Material and methods

### Animals

The SPF (Beijing) Biotechnology Co., Ltd. provided the female BALB/c mice that were used in this experiment. All mice were maintained in clean, pathogen-free animal cabins with unrestricted access to standard laboratory food and water, on a 12-hour light/dark cycle, and in an environment with regulated humidity (60%-80%) and temperature (22±1˚C). Tianjin Medical University's Institute Animal Treatment and Use Committee (IACUC) accepted the animal experiment, which was carried out strictly in compliance with institutional ethical standards for the care of animals,and the Guide for the Care and Use of Laboratory Animals by certified staff in an Association for Assessment and Accreditation of Laboratory Animal Care (AAA-LAC) International accredited facility. Animals were anesthetized using isoflurane (RWD Life science Co., LTD, Shenzhen, China) and sacrificed using $CO_2$ and cervical dislocation per AAALAC guidelines. All animals were monitored for signs of suffering and, if any had been present, mice would have euthanized to avoid prolonged suffering.

### Reagents and treatments

Calcipotriol (Cal) ointment was provided by LEO Pharmaceutical (0.005%; LEO Pharmaceutical, Denmark), Imiquimod (IMQ) cream was provided by Mingxin Pharmaceutical (0.5%; Mingxin Pharmaceutical, SiChuan, China); Medical petroleum jelly was provided by Qingdao Hainuo Group (Qingdao Hainuo Group, QingDao, China); Glycyrrhizic acid (Sigma-Aldrich, Germany), Astilbin (Sigma-Aldrich, Germany), Osthole(Sigma-Aldrich, Germany), Momordin Ic (Sigma-Aldrich, Germany). The production of compound ointment: heat and melt the petroleum jelly first, then gradually add the medicine in stages while stirring continuously until it condenses. The drug content in compound ointment was Glycyrrhizic acid (1.5%), Astilbin(0.05%), Osthole(0.05%), and Momordin Ic(0.05%).

Eight-week-old female BALB/c mice were randomly divided into 6 groups (n = 10 per group): control, IMQ (IMQ-applied only), IMQ + Cal (IMQ with Calcipotriol Ointment), IMQ + GA (50 mg/d), IMQ + GA (100 mg/d), and IMQ + GA (200 mg/d). After the mice were acclimatized for one week, the hair on the back was completely removed (2 cm × 3 cm) using hair removal cream. Mice in the drug administration group had the corresponding drug applied daily on the back with a glass rod and wiped the drug off after 1 h using a sterile cotton swab. In contrast, the control and model groups only applied 0.2 g of medical petroleum jelly.

After the mice were given the drug for 1h and wiped clean, the psoriasis model was prepared by applying 5% imiquimod cream 62.5 mg on the back.

## Scoring severity of skin inflammation

To assess the degree of skin inflammation on the back, an objective scoring system based on the clinical Psoriasis Area and Severity Index (PASI) was developed. Erythema, scaling, and thickening were all rated individually on a scale of 0 to 4, as follows: 0, none; 1, faint; 2, moderate; 3, marked; 4, extremely marked. The intensity of inflammation was evaluated by the cumulative score (erythema plus scaling plus thickness, scale 0–12).

## Spleen index

Before being sacrificed with isoflurane(RWD Co., Ltd, Shengzhen, China), the mice were weighed, and then each spleen was weighed. To capture the dynamic body changes of the mice, we calculated the spleen index (spleen index-spleen weight/body weight).

## Histopathological of psoriasis in mice

On the eighth day of the experiment, shaved back skin samples from each group of mice were fixed in 4% formaldehyde, embedded in paraffin, and cut into 4-μm sections. Hematoxylin and eosin (H&E) were used to stain the sections.

All fields of view of each H&E-stained section were photographed continuously (200× magnification) using natural light sources with manual exposure. H&E sections were analyzed using Image Pro-plus6.0 software (IPP, Media Cybernetics, USA). Calibration of optical density and scale bar. Echinoderm thickness measurement: the anterior, middle and posterior fields of view of each section were measured, and the thickness of the echinoderm was measured at five different random locations in each field of view.

## Cytokine concentration in skin tissue

The mice were sacrificed on day 8, and 50mg of skin was taken, cut up, and added to 500μL of cell lysate and put into a tissue mill for 30min, then centrifuged at 3000rpm for 20min at room temperature. Finally, the contents of IL-12, IL-17A, IL-23, and TNF-α in the supernatant were measured according to the instructions of the Enzyme-linked immunosorbent assay (ELISA) kit.

## Statistical analysis

GraphPad Prism 8.0 statistical software was used for the statistical analysis. Except for the study of clinical ratings, which utilized two-way ANOVA followed by Dunnett-t test, the experiment involving several groups employed a one-way analysis of variance (ANOVA) followed by Turkey's multiple comparison test. P values of less than 0.05 were regarded as significant.

# Results

## Glycyrrhizic acid compound ointment dramatically reduced IMQ-induced psoriasis-like inflammation in mice

We investigated the effect of different doses of glycopyrrolate compound ointment on psoriasis-like inflammation in psoriasis mouse model induced by IMQ. The dorsal skin of BALB/c mice was treated with IMQ cream for 7 days, with or without calcipotriol ointment or

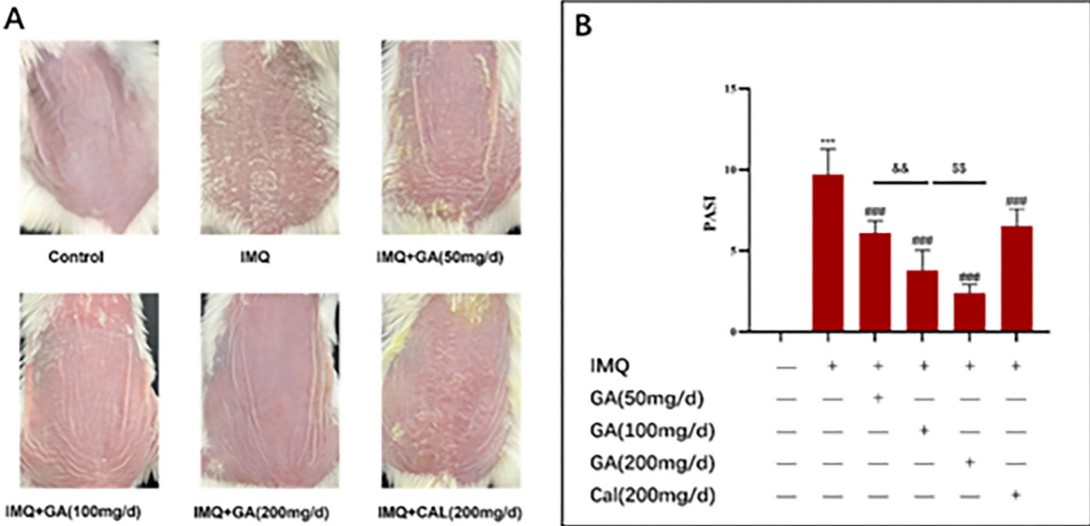

**Fig 1. Glycyrrhetinic acid ointment inhibited IMQ-induced psoriasis-like inflammation.** (A) Phenotypic observation of the dorsal skin of the treated mice group. (B) The clinical score of imiquimod-induced psoriasiform dermatitis mice (Mean± SD, N = 10). IMQ, Imiquimod; GA, Glycyrrhetinic acid compound ointment; Cal, Calcipotriol. ***$P<0.01$ vs normal group; ###$P<0.001$ vs model group; &&$P<0.01$ vs GA(200 mg/d) group;$^{\$\$}$$P<0.01$ vs GA(100 mg/d) group.

glycopyrrolate compound ointment. As shown in Fig 1, after IMQ treatment, typical psoriasis-like inflammation, such as erythema, scaling, and thickening, was observed compared with control group. All glycopyrrolate compound ointment and calcipotriol groups show a lower PASI, in which glycopyrrolate compound ointment (50mg/d) was equivalent to calcipotriol (200mg/d). Glycyrrhizic acid compound ointment high dose (200 mg/d) and medium doses (100 mg/d) doses group scores significantly lower than IMQ model group (Fig 1). The data showed that glycyrrhizic acid compound ointment was dose-dependent and more effective than calcipotriol ointment in treating IMQ-induced psoriasis-like dermatitis in mice. Glycyrrhizic acid compound ointment significantly reduced psoriasis-like inflammation, as evidenced by the reduction in the severity score of dorsal skin inflammation, which was better than calcipotriol, a vitamin D analogue for psoriasis.

## Glycyrrhizic acid compound ointment reduced splenomegaly in IMQ-treated mice

Previous study found that IMQ caused splenomegaly via systemic effects [10]. IMQ was applied locally to mice for 7 consecutive days, and the spleen was taken. The spleen was smallest in the high dose group (200 mg/d) of glycyrrhizic acid compound ointment, followed by the medium dose (100 mg/d) and low dose (50 mg/d), but there was little change in the spleen size in the calcipotriol ointment group (Fig 2A). This finding suggested that glycyrrhizic acid compound ointment affected IMQ-induced splenomegaly. The spleen index was calculated for each group. The splenic index of glycyrrhizic acid compound ointmen high-dose group (200 mg/d) was significantly lower medium-dose (100 mg/d) and low-dose (50 mg/d) groups. The above indices were also significantly lower in the calcipotriol ointment group than the IMQ group (P<0.01, Fig 2B). The above data demonstrated that glycyrrhizic acid compound ointment also showed a dose-dependent effect on IMQ-induced splenomegaly and a more substantial therapeutic effect than calcipotriol ointment.

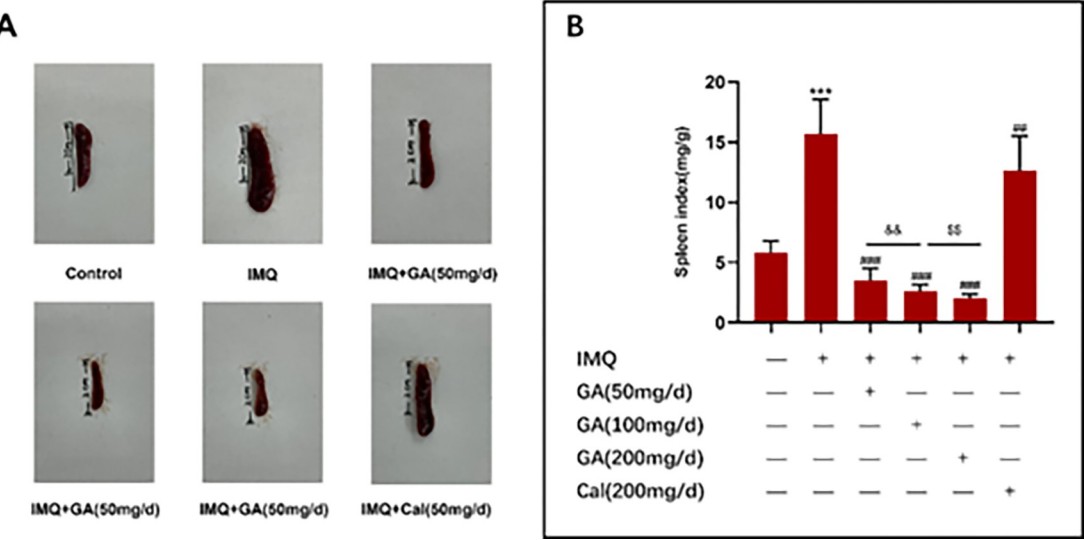

**Fig 2. The spleen index of mice with imiquimod-induced psoriasiform dermatitis.** (A) Representative spleen pictures from each group. (B) Spleen weight to body weight ratios (spleen index) (Mean SD, N = 10). ***$P < 0.01$ vs control group; ###$P < 0.001$ vs model group; &&$P < 0.01$ vs GA (200 mg/d) group; $$P < 0.01$ vs GA(100 mg/d) group.

## H&E section staining

The H&E results showed that the histological changes in the IMQ group exhibited extensive psoriasis-like lesions. Furthermore, it has been observed that the mice in the IMQ group had obvious hyperkeratosis, underkeratosis, mild echinoderm, extended reticular ridges, and mild dermal inflammation. In comparison, the hyperkeratosis and underkeratosis in both high and medium-dose groups of glycyrrhizic acid compound ointment were significantly reduced compared with the model group, in which the high-dose group were most apparent (Fig 3A).

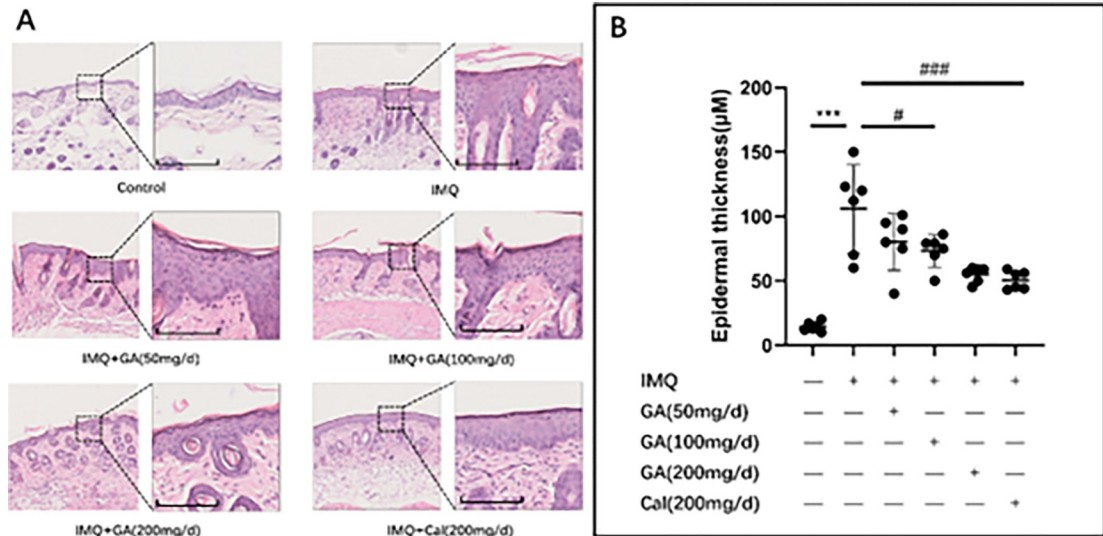

**Fig 3. The H&E section staining and epidermal thickness of imiquimod-induced psoriasiform.** (A) H&E staining of the back skin of mice from each group. Scale bar, 100μm. (B) Epidermal thickness was evaluated.dermatitis mice. ***$P < 0.001$, #$P < 0.05$, ###$P < 0.001$.

The thickness of the spine layer of the high-dose group was decreased compare to IMQ group, similar to the calcipotriol group (Fig 3B). However, the thickness of the spine layer in the low-dose group was not significantly different from that in the IMQ group. The above results indicated that glycyrrhizic acid compound ointment could reduce the phenomenon of incomplete and hyperkeratosis caused by psoriasis and, simultaneously, the thickness of the spiny layer.

## Glycyrrhetinic acid compound ointment treatment reduced the expression of pro-inflammatory cytokines in IMQ-treated skin

The IL-23/IL-17 axis was involved in IMQ-induced psoriasis-like skin inflammation in mice. Dendritic cell-produced IL-23 stimulated Th17 cell proliferation, which activated and secreted IL-17A, TNF-α, IL-22, and IL-12 induced Th cell differentiation to Th1 cells [7,9,10]. The expression of these four inflammatory cytokines in the skin of the IMQ group was significantly higher than in the control group. As shown in Fig 4, these cytokines, IL-17A, TNF-α, IL-22, and IL-12, were inhibited dose-dependently by glycyrrhizic acid compound ointment. The inhibition effect of calcipotriol ointment was less than the glycyrrhizic acid compound ointment. The above data suggested that glycyrrhizic acid ointment could effectively reduce inflammatory factors and attenuate the IMQ-induced inflammatory response.

## Discussion

According to previous research, epidermal hyperproliferation, premature maturation of keratin-forming cells, incomplete keratinization, and retention of the nucleus in the stratum corneum are the critical characteristics of psoriasis [11]. The imbalance between keratin-forming cell proliferation and differentiation leads to epidermal overproliferation that exceeds the degree of skin damage [12]. Therefore, inhibition of keratinocyte overproliferation has long been considered as a potential strategy for treating psoriasis [13,14].

In vitro studies revealed that GA reduced HaCaT keratinocyte activity and induced apoptosis and that GA-mediated apoptosis of HaCaT keratinocytes was linked to induction of ROS and inhibition of the PI3K-AKT signaling pathway [5]. We found that glycyrrhizic acid compound ointment significantly improved psoriatic development. In addition, GA was reported to ameliorate psoriasis symptoms by inhibiting inflammatory cytokines and the STAT3/mTOR signaling pathway [15]. It had been shown that a significant bioactive compound, astilbin, extracted from *Smilax glabra* Roxb, had anti-keratin-forming cell overproliferation and cell cycle arrest effects on HaCaT cells and had keratin-forming cell differentiation regulatory

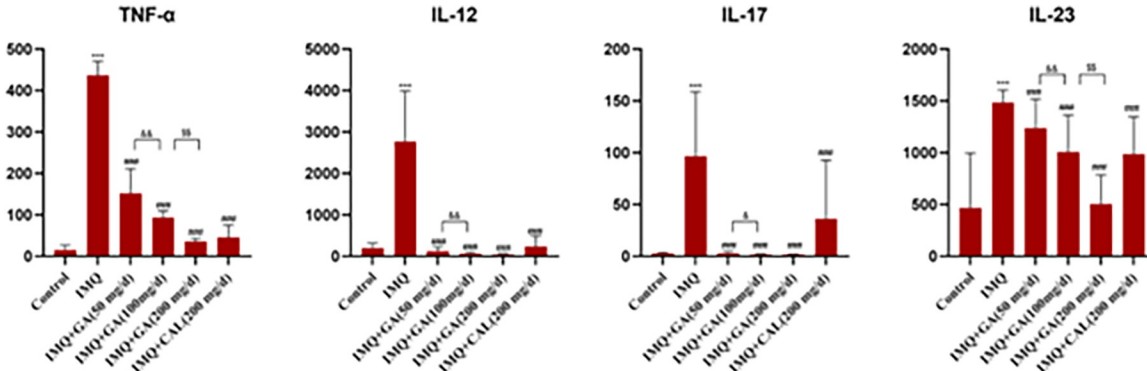

**Fig 4. The skin levels of TNF-α, IL-12, IL-17A, and IL-23 in imiquimod-induced psoriasiform dermatitis mice (pg/ml, Mean± SD, N = 10).** \*\*\*$P<0.01$ *vs* normal group; \#\#\#$P<0.001$ *vs* model group; &&$P<0.01$ *vs* GA(200 mg/d) group;\$\$$P<0.01$ *vs* GA(100 mg/d) group.

activity [16]. The ability of osthole to induce HaCaT cells apoptosis in a dose-dependent manner suggested that osthole may exert antipsoriatic effects by inhibiting epidermal cell overproliferation and inducing epidermal cell apoptosis [17]. The inhibitory effect of momordin Ic in *Kochia scoparia* (L.) Schrad. on HaCaT cell proliferation and the alteration of the Wnt/β-linked protein pathway could be induced by momordin Ic [18]. We hypothesized that glycyrrhizic acid complex ointment reduced PASI scores in murine psoriasis-like lesions through its inhibitory effect on excessive proliferation of keratin-forming cells. Interestingly, in this research we found PASI score in the calcipotriol ointment-treated group was comparable to the glycyrrhizic acid compound ointment low-dose group. However, the HE staining showed that pathological damage in the calcipotriol ointment-treated group was comparable to the high-dose group, suggesting that the PASI score could not fully express the degree of psoriasis lesions in mice.

Therefore, other indicators are needed to combine the extent of psoriasis lesions in mice, such as histological observations and pro-inflammatory factors. The spleen is an essential part of the immune system. Splenomegaly caused by IMQ is linked to its systemic effects [19], while glycyrrhizic acid compound ointment significantly reduced the splenic index. Presumably, glycyrrhizic acid compound ointment had a solid immunosuppressive effect. However, the splenic index was not significantly reduced in the calcipotriol ointment treatment group. Recent studies had shown that 4nM calcipotriol treatment did not significantly increase the frequency of splenic B cells and is capable of causing atopic dermatitis [20]. We speculated that the modest effect of calcipotriol on splenomegaly is due to the high dose, which warrants further investigation.

It is currently hypothesized that the key to the pathogenesis of psoriasis is the activation of T cells [21], which release a variety of cytokines and chemokines that interact with keratin-forming cells and other immune cells (dendritic cells, Langerhans cells, neutrophils) to form a complex cytokine network that mediates and maintains the development of psoriatic inflammation (Fig 5) [22]. Th1 and Th17 cells of the helper T-cell subsets are currently considered the most closely related to the pathogenesis of psoriasis [23]. Th1 cells secrete IFN-γ, IL-12, and TNF-α to generate cellular immunity, and IL-12 induces differentiation of Th cells to Th1 cells [24]. Th17 cells and their secreted cytokines, such as IL-17A, IL-23, and TNF-α, play a role in a variety of chronic inflammatory illnesses, including psoriasis [10]. These Th17-associated cytokines can aid in the differentiation of Th17 cells. These Th17-related cytokines can also have an effect on keratinocytes and other inflammatory cells in the skin, causing local inflammation and keratinocyte proliferation [21]. All these findings confirm the role of IL-23 in the development of skin diseases. Previous studies have shown anti-inflammatory activity of Glycyrrhizic acid, astilbin, osthole, and momordin Ic [5,17,18,25]. The data in our study showed that glycyrrhizic acid compound ointment inhibited inflammatory cytokines including TNF-α, IL-12, IL-17A, and IL-23 in the skin of IMQ-induced psoriasis-like mice, and the inhibitory effect became stronger with increasing dose. Interestingly, Glycyrrhizic acid compound ointment inhibited IL-12 and IL-17A significantly, which we speculate may have reduced the expression of IL-17A and IL-23 by inhibiting the activation process of Th17 cells. Calcipotriol, a low calcium analog of vitamin D3, stimulated high expression of TSLP in mouse epithelial keratinocytes, which induces DC activation and promotes the differentiation of CD4$^+$ T cells to Th2 cells [26], followed by Th2-type inflammation, causing atopic dermatitis syndrome in mice. Calcipotrioldid did not have a strong inhibitory effect on IL-23/IL-17A axis-induced inflammation, which was consistent with our finding that Calcipotriol did not significantly inhibit IL-17A and IL-23, demonstrating that Calcipotriol was not suitable for the treatment of IMQ-induced psoriasis-like mice.

Comorbidities associated with psoriasis include cardiovascular disease and metabolic syndrome. There is no cure for this condition due to its chronic nature and the high chance of

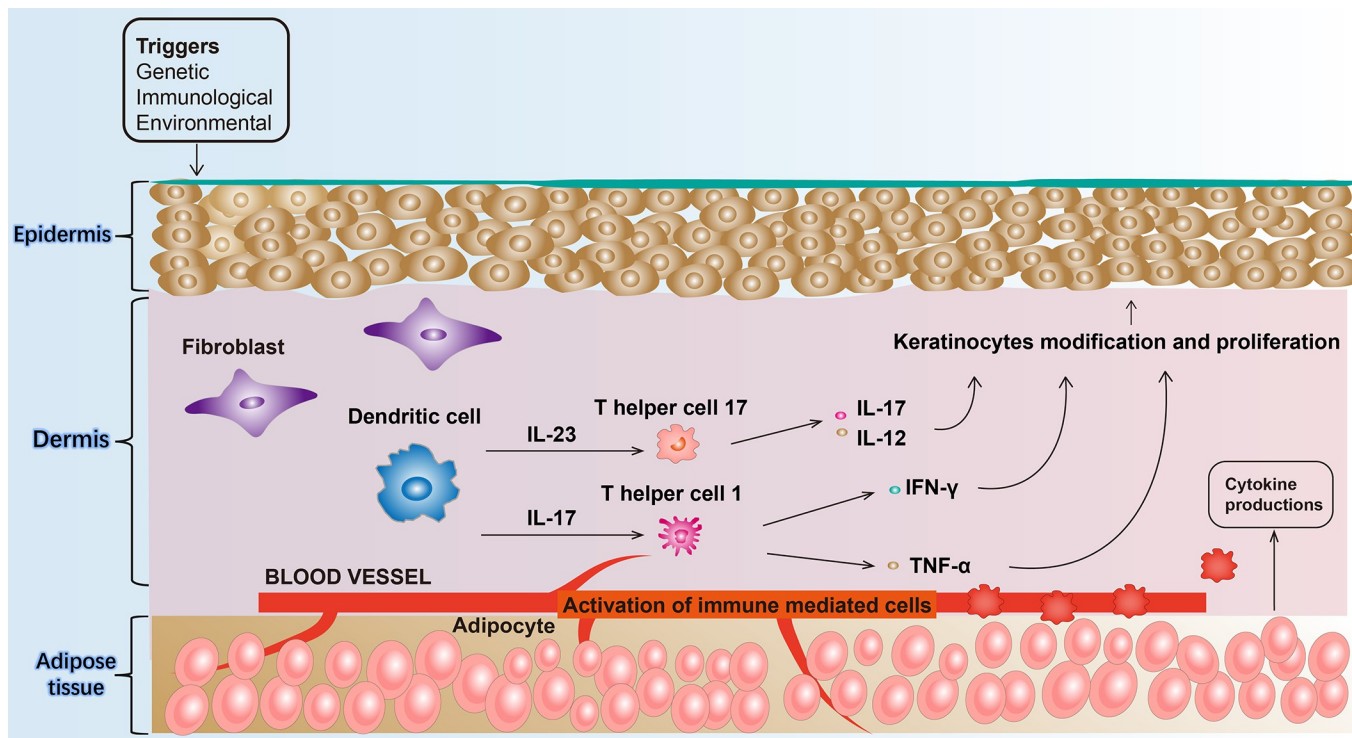

**Fig 5. Pathophysiology of psoriasis.**

recurrence [27]. Although glucocorticoids, calcium, and retinoids are effective in reducing psoriasis, none of them can eradicate psoriasis. In addition, many drugs can cause serious side effects, such as immune disorders, hematopoietic dysfunction, skin atrophy, and capillary dilation [28]. The treatment of psoriasis involves controlling the disease, slowing the progression, and reducing the self-conscious symptoms and skin damage. Over the past few years, many treatments for psoriasis have been generated, and herbal remedies are one of the emerging approaches. Many herbal monomer have been shown to have antipsoriatic effects [29]. Among them, we found that a compound ointment with glycyrrhizic acid as the main ingredient and a variety of other herbal monomer as supplements can reduce psoriasis symptoms and is superior to Calcipotriol in skin performance. Glycyrrhizic acid combination ointment may be used as an adjunct to chemotherapy in the treatment of psoriasis, and the combination may be able to reduce the dose of chemotherapy and mitigate adverse effects.

## Conclusions

Glycyrrhetinic acid compound ointment was more effective than calcipotriol and was dose-dependent in the treatment of imiquimod-induced psoriatic dermatitis in mice. Meanwhile, calcipotriol was not suitable for the treatment of IMQ-induced psoriasis-like mice.

## Acknowledgments

The authors are grateful to Tianjin Medical University for the provision of experimental sites.

## Author Contributions

**Conceptualization:** Shuangyong Sun.

**Data curation:** Yanwen Zhang.

**Formal analysis:** Yanwen Zhang.

**Methodology:** Yanwen Zhang.

**Project administration:** Lingyan Jiang.

**Resources:** Qian Wang.

**Software:** Qian Wang.

**Supervision:** Qian Wang.

**Validation:** Shuangyong Sun.

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
