## [Decision Letter · Decision Letter 0]

9 May 2023

PONE-D-22-29974The therapeutic effect of glycyrrhizic acid compound ointment on imiquimod-induced psoriasis-like disease in micePLOS ONE

Dear Dr. Sun,

Thank you for submitting your manuscript to PLOS ONE. After careful consideration, we feel that it has merit but does not fully meet PLOS ONE’s publication criteria as it currently stands. Therefore, we invite you to submit a revised version of the manuscript that addresses the points raised during the review process.

We look forward to receiving your revised manuscript.

Kind regards,

Jianhong Zhou

Staff Editor

PLOS ONE

Journal Requirements:

2. As part of your revision, please complete and submit a copy of the Full ARRIVE 2.0 Guidelines checklist, a document that aims to improve experimental reporting and reproducibility of animal studies for purposes of post-publication data analysis and reproducibility: https://arriveguidelines.org/sites/arrive/files/documents/Author%20Checklist%20-%20Full.pdf (PDF). Please include your completed checklist as a Supporting Information file. Note that if your paper is accepted for publication, this checklist will be published as part of your article.

3.  To comply with PLOS ONE submissions requirements, in your Methods section, please provide additional information regarding the experiments involving animals and ensure you have included details on (1) methods of sacrifice, (2) methods of anesthesia and/or analgesia, and (3) efforts to alleviate suffering.

   "No"

  "The authors are grateful to Tianjin Medical University for the provision of experimental sites.This research was funded by the National Natural Science Foundation of China (Grant No. 32170110）"

  "No"

7. Thank you for stating the following in your Competing Interests section:  

   "NO authors have competing interests"

8. PLOS requires an ORCID iD for the corresponding author in Editorial Manager on papers submitted after December 6th, 2016. Please ensure that you have an ORCID iD and that it is validated in Editorial Manager. To do this, go to ‘Update my Information’ (in the upper left-hand corner of the main menu), and click on the Fetch/Validate link next to the ORCID field. This will take you to the ORCID site and allow you to create a new iD or authenticate a pre-existing iD in Editorial Manager. Please see the following video for instructions on linking an ORCID iD to your Editorial Manager account: https://www.youtube.com/watch?v=_xcclfuvtxQ

9. In your Data Availability statement, you have not specified where the minimal data set underlying the results described in your manuscript can be found. PLOS defines a study's minimal data set as the underlying data used to reach the conclusions drawn in the manuscript and any additional data required to replicate the reported study findings in their entirety. All PLOS journals require that the minimal data set be made fully available. For more information about our data policy, please see http://journals.plos.org/plosone/s/data-availability.

Additional Editor Comments:

Dear Sir

Kindlyrevise this ms as suggested by reviewers.

Reviewers' comments:

Reviewer's Responses to Questions

**Comments to the Author**

1. Is the manuscript technically sound, and do the data support the conclusions?

Reviewer #1: Yes

Reviewer #2: Yes

2. Has the statistical analysis been performed appropriately and rigorously? 

Reviewer #1: Yes

Reviewer #2: Yes

3. Have the authors made all data underlying the findings in their manuscript fully available?

Reviewer #1: Yes

Reviewer #2: Yes

4. Is the manuscript presented in an intelligible fashion and written in standard English?

Reviewer #1: Yes

Reviewer #2: Yes

5. Review Comments to the Author

Reviewer #1: New, safer treatments for psoriasis are needed. This manuscript provides a new approach that could also be safe and readily available. The experiments are well done, statistically valid and potentially clinically important. The English can be improved.

Reviewer #2: The current study represents a clear story investigating the potential therapeutic effects of glycyrrhizin on psoriasis induced in mice. Psoriasis-like mouse-model was chosen carefully, and experiments were carried out elegantly to support their idea. However, a quick literature review {34044769; 27786567; 26629143} showed studies presenting similar idea that glycyrrhizin improves psoriasis. I believe it’s crucial the authors need to more clearly explain the novel addition of their study to the knowledge we already have regarding glycyrrhizin therapeutic effects.

The study was carried out on female mice only. Is there a specific reason for that? Are male mice going to be studied later on?

In the H&E experiments, fig.3, How was the epidermal thickness determined? How were such results normalized with regard to normalization pixelated images, number of samples/sections per mice?

In fig.5, the figure is a great addition to the article, an explanatory legend would be beneficial.

In the discussion section, it was stated that the current hypothesis for psoriasis pathogenesis is through the activation of T cells. I believe more references are needed to support this hypothesis.

Minor comments:

Statement of funding is not complete

In the discussion section: paragraph 2, first line: check for spelling

6. PLOS authors have the option to publish the peer review history of their article (what does this mean?). If published, this will include your full peer review and any attached files.

Reviewer #1: **Yes: **James David Adams Jr PhD

Reviewer #2: No

---

## [Author Response · Author response to Decision Letter 0]

28 Jun 2023

Reviewer #1

Comment: New, safer treatments for psoriasis are needed. This manuscript provides a new approach that could also be safe and readily available. The experiments are well done, statistically valid and potentially clinically important. The English can be improved.

Response: Thank you for this valuable feedback, and the whole manuscript has been polished accordingly.

Reviewer #2

Comment 1: However, a quick literature review {34044769; 27786567; 26629143} showed studies presenting similar idea that glycyrrhizin improves psoriasis. I believe it’s crucial the authors need to more clearly explain the novel addition of their study to the knowledge we already have regarding glycyrrhizin therapeutic effects.

Response: Although the antipsoriatic effects of glycyrrhizic acid have been similarly reported, our paper was optimized for the associative properties of glycyrrhizic acid and other natural drug monomers. It was the most potent natural drug monomer blend ointment reported to date, and it achieved the potency of calcipotriol, which had not been demonstrated by any other method before. And unlike other studies, we used transdermal administration rather than oral administration, which significantly reduced the dose administered. It was an important addition to glycopyrrolate for psoriasis treatment.

At the same time, we found no reduction in spleen weight in psoriatic mice with calcipotriol ointment. It was reported that calcipotriol induced atopic dermatitis-like lesions in mice, so we questioned the use of calcipotriol as a positive drug in a mouse psoriasis model. We show this experimental phenomenon of calcipotriol as a positive drug when studying psoriasis and hope that other investigators will take this into account in future studies.

We thank the reviewer for this comment and realize that these differences may not have been expressed clearly enough in the previous manuscript. We have improved the original manuscript, particularly in the introduction (paragraph 2), to clearly illustrate the differences in this study. 

Comment 2: The study was carried out on female mice only. Is there a specific reason for that? Are male mice going to be studied later on?

Response: Thanks for reviewer’s comment, the study by Pilar Alvarez et al. [1] showed that Female mice develop more severe disease than male mice in response to generic imiquimod cream. Meanwhile, in the course of our experiments, we found that male mice induced with imiquimod showed mostly no skin damage. With regard to gender differences in psoriasis models, this is something we need to work on in the future and we are working on this, and you will see research in this area in our future work.

[1]Pilar Alvarez, Liselotte E. Jensen, "Imiquimod Treatment Causes Systemic Disease in Mice Resembling Generalized Pustular Psoriasis in an IL-1 and IL-36 Dependent Manner", Mediators of Inflammation, vol. 2016, Article ID 6756138, 10 pages, 2016. https://doi.org/10.1155/2016/6756138

Comment 3: In the H&E experiments, fig.3, How was the epidermal thickness determined? How were such results normalized with regard to normalization pixelated images, number of samples/sections per mice? 

Response: We apologize for the omission of the description of the skin measurement method, the relevant method has been added to the Materials and Methods section:

“All fields of view of each H&E-stained section were photographed continuously (200× magnification) using natural light sources with manual exposure. HE sections were analyzed using Image Pro-plus6.0 software (IPP, Media Cybernetics, USA). Calibration of optical density and scale bar. Echinoderm thickness measurement: the anterior, middle and posterior fields of view of each section were measured, and the thickness of the echinoderm was measured at five different random locations in each field of view.”

Comment 4: In the discussion section, it was stated that the current hypothesis for psoriasis pathogenesis is through the activation of T cells. I believe more references are needed to support this hypothesis.

Response: We are very sorry for our careless mistake and we have added references in the discussion section:

“[21]Coimbra S, Figueiredo A, Castro E, Rocha-Pereira P, Santos-Silva A. The roles of cells and cytokines in the pathogenesis of psoriasis. Int J Dermatol. 2012;51: 389–398. doi:10.1111/j.1365-4632.2011.05154.x”

Comment 5: Statement of funding is not complete. In the discussion section: paragraph 2, first line: check for spelling.

Response: We are very sorry for our careless mistake and we have fixed the spelling mistake of "reduced". Meanwhile, we have added the fund information to the online submission form.

---

## [Decision Letter · Decision Letter 1]

13 Aug 2023

The therapeutic effect of glycyrrhizic acid compound ointment on imiquimod-induced psoriasis-like disease in mice

PONE-D-22-29974R1

Dear Dr. Shuangyong Sun,

We’re pleased to inform you that your manuscript has been judged scientifically suitable for publication and will be formally accepted for publication once it meets all outstanding technical requirements.

Kind regards,

Masanori A. Murayama

Academic Editor

PLOS ONE

Additional Editor Comments (optional):

Thank you for submitting revised manuscript. In this time I will announce the decision as Accept. Congratulations.

Reviewers' comments:

Reviewer's Responses to Questions

**Comments to the Author**

1. If the authors have adequately addressed your comments raised in a previous round of review and you feel that this manuscript is now acceptable for publication, you may indicate that here to bypass the “Comments to the Author” section, enter your conflict of interest statement in the “Confidential to Editor” section, and submit your "Accept" recommendation.

Reviewer #1: All comments have been addressed

Reviewer #2: All comments have been addressed

2. Is the manuscript technically sound, and do the data support the conclusions?

Reviewer #1: Yes

Reviewer #2: Yes

3. Has the statistical analysis been performed appropriately and rigorously? 

Reviewer #1: Yes

Reviewer #2: Yes

4. Have the authors made all data underlying the findings in their manuscript fully available?

Reviewer #1: Yes

Reviewer #2: Yes

5. Is the manuscript presented in an intelligible fashion and written in standard English?

Reviewer #1: Yes

Reviewer #2: Yes

6. Review Comments to the Author

Reviewer #1: The manuscript has been adequately revised and is ready to publish without revision. The English is adequate.

Reviewer #2: Thank you for taking the time to address each comment thoroughly and making the corresponding changes.

7. PLOS authors have the option to publish the peer review history of their article (what does this mean?). If published, this will include your full peer review and any attached files.

Reviewer #1: **Yes: **James David Adams Jr

Reviewer #2: No

---

## [Editor Report · Acceptance letter]

16 Aug 2023

PONE-D-22-29974R1 

The therapeutic effect of glycyrrhizic acid compound ointment on imiquimod-induced psoriasis-like disease in mice 

Dear Dr. Sun:

I'm pleased to inform you that your manuscript has been deemed suitable for publication in PLOS ONE. Congratulations! Your manuscript is now with our production department. 

Kind regards, 

on behalf of

Dr. Masanori A. Murayama 

Academic Editor

PLOS ONE